# Comparative genomics analysis of three conserved plasmid families in the Western Hemisphere soft tick-borne relapsing fever borreliae provides insight into variation in genome structure and antigenic variation systems

Alexander R. Kneubehl,[1] Job E. Lopez[1,2]

**ABSTRACT** *Borrelia* spirochetes, causative agents of Lyme disease and relapsing fever (RF), have uniquely complex genomes, consisting of a linear chromosome and both circular and linear plasmids. The plasmids harbor genes important for the vector-host life cycle of these tick-borne bacteria. The role of plasmids from Lyme disease causing spirochetes is more refined compared to RF *Borrelia* because of limited plasmid-resolved genome assemblies for the latter. We recently addressed this limitation and found that three linear plasmid families (F6, F27, and F28) were syntenic across all the RF *Borrelia* species that we examined. Given this conservation, we further investigated the three plasmid families. The F6 family, also known as the megaplasmid, contained regions of repetitive DNA. The F27 was the smallest, encoding genes with unknown function. The F28 family encoded the putative expression locus for antigenic variation in all species except *Borrelia hermsii* and *Borrelia anserina*. Taken together, this work provides a foundation for future investigations to identify essential plasmid-localized genes that drive the vector-host life cycle of RF *Borrelia*.

**IMPORTANCE** *Borrelia* spp. spirochetes are arthropod-borne bacteria found globally that infect humans and other vertebrates. RF borreliae are understudied and misdiagnosed pathogens because of the vague clinical presentation of disease and the elusive feeding behavior of argasid ticks. Consequently, genomics resources for RF spirochetes have been limited. Analyses of *Borrelia* plasmids have been challenging because they are often highly fragmented and unassembled in most available genome assemblies. By utilizing Oxford Nanopore Technologies, we recently generated plasmid-resolved genome assemblies for seven *Borrelia* spp. found in the Western Hemisphere. This current study is an in-depth investigation into the linear plasmids that were conserved and syntenic across species. We identified differences in genome structure and, importantly, in antigenic variation systems between species. This work is an important step in identifying crucial plasmid-localized genetic elements essential for the life cycle of RF spirochetes.

**KEYWORDS** *Borrelia*, relapsing fever, bioinformatics, adaptive sampling

Address correspondence to Job E. Lopez, job.lopez@bcm.edu.

The authors declare no conflict of interest.

See the funding table on p. 20.

The plasmid content of the *Borreliaceae* is the most unique and complex among bacteria (1–5). No other known bacterial organism harbors such a diverse repertoire of linear and circular plasmids, which are necessary for the completion of the microbes' infectious cycle through the arthropod vector and vertebrate host (6–18). Extensive research has been conducted on the function of plasmids in *Borreliella* (*Borrelia*) *burgdorferi* (3–6, 19–22). This has proven important in the delineation of inter- and

intra-species plasmid relationships and the identification of essential genes. Comparatively less work has been performed in tick-borne relapsing fever (RF) spirochetes because plasmid-resolved genome assemblies for these microbes have been lacking.

We previously reported a comparative genomic analysis of seven species of Western Hemisphere soft tick-borne RF (WHsTBRF) spirochetes (1). A phylogenetic analysis of the PF32 plasmid partitioning loci identified 30 different plasmid families (F1–F30). Of these, three (F6, F27, and F28) were conserved and largely syntenic across the clade. The F6 plasmid family, also known as the megaplasmid, was the largest linear plasmid (110–194 kb). The F27 plasmid was the smallest (10–12 kb) encoding 12–14 genes of unknown function. The F28 plasmid family was related to the *B. burgdorferi* cp26 plasmid, which is essential in the life cycle of Lyme disease (LD)-causing spirochetes.

In this current study, we performed a comparative analysis of the F6, F27, and F28 plasmid families across the WHsTBRF spirochete clade. The species that we evaluated were *Borrelia anserina*, *Borrelia coriaceae*, *Borrelia hermsii*, *Borrelia parkeri*, *Borrelia puertoricensis*, *Borrelia turicatae*, and *Borrelia venezuelensis*. This investigation revealed extensive differences in repetitive gene content across the F6 megaplasmid family. Moreover, the F27 plasmid family was determined to be larger than previously reported due to an ancestral inverted duplication event. Interestingly, the F28 plasmid family contained the expression site for antigenic variation for all species except *B. hermsii* and *B. anserina*. This finding led to further investigation of the antigenic variation systems across the WHsTBRF spirochete clade. Collectively, this work is foundational for studies that will investigate the role of plasmid families in RF spirochete vector colonization, host use, vector specificity, and pathogenesis.

## RESULTS

### F6 (megaplasmid) plasmid family analysis

As the largest, non-chromosomal linear replicon found in RF spirochete genomes (23), the overall length of the F6 plasmid family varied. The largest megaplasmid was found in *B. hermsii* YOR at 194 kb, and the smallest was found in *B. anserina* BA2 at 88 kb. An analysis of the relatedness of WHsTBRF spirochetes' megaplasmids by Mauve alignment indicated variable sequence conservation (Fig. 1). We identified three regions that were similar to those reported by Miller et al. (23). Starting from the 5′ end of the megaplasmid, region 1 was variable in length between all the species and contained *bbk32* genes, which encode fibronectin-binding proteins, and multiple gene copies encoding complement regulator-acquiring surface proteins (CRASPs). Region 2 contained plasmid partitioning genes, as well as *nrdF*, *nrdE*, and *nrdI*, which encode ribonucleoside-diphosphate reductase 2 subunit beta, ribonucleoside-diphosphate reductase 2 subunit alpha, and ribonucleotide reductase Class Ib proteins, respectively. A gene encoding *Borrelia* immunogenic protein A (*bipA*), a diagnostic antigen for tick-borne RF (TBRF) spirochetes (24–26), was found in region 2 of all isolates except for *B. anserina* BA2, which did not encode a homolog. Intact phage-related genes were also found in region 2 [e.g., genes encoding PBSX family phage terminase large subunit, multi-copy lipoprotein (Mlp) family, and BlyA family holin proteins]. Region 3 was variable in length and gene content between all species and contained repetitive nucleotide sequences. Overall, the three regions of conserved gene content varied in length across the WHsTBRF spirochete megaplasmids.

The variation in the length of the megaplasmids was primarily due to repetitive nucleotide sequences. This was reported in *B. hermsii* where five areas of genes with a high degree of repetitive nucleotide sequence were identified (23). We applied the designations of repetitive nucleotide blocks A–E across the WHsTBRF clade (Fig. 1). The genes found in blocks A–E are indicated in Supplementary file 1 with their InterProScan classifications. Repetitive blocks A and B were found in what was designated as region 1, block C was in region 2, and blocks D and E were in region 3 (annotated in Fig. 1). Self-dot plot analysis revealed the presence of these repetitive blocks was variable across the WHsTBRF clade (Fig. 2).

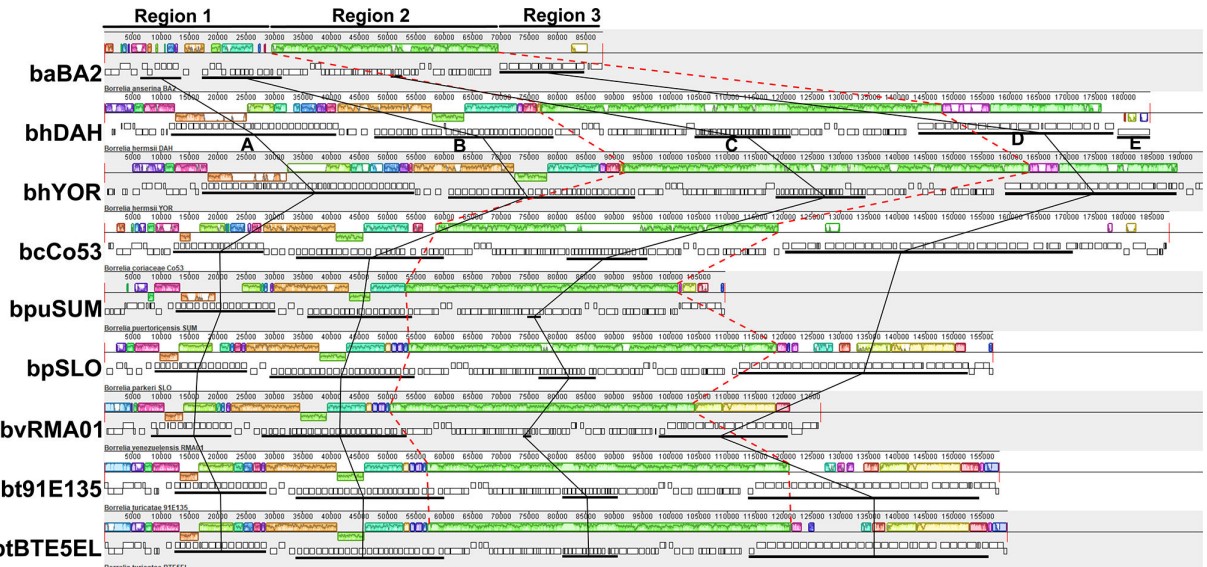

**FIG 1** Mauve alignment of WHsTBRF megaplasmids (F6 Family). Three regions of interest are highlighted, which correspond to areas of variability and conservation. Strain designations are indicated on the left side of each plasmid (baBA2: *B. anserina* BA2, bhDAH: *B. hermsii* DAH, bhYOR: *B. hermsii* YOR, bcCo53: *B. coriaceae* Co53, bpuSUM: *B. puertoricensis* SUM, bpSLO: *B. parkeri* SLO, bvRMA01: *B. venezuelensis* RMA01, bt91E135: *B. turicatae* 91E135, and btBTE5EL: *B. turicatae* BTE5EL). Red dashed lines are used to show where region 2 starts and stops for each isolate. The repetitive regions previously defined in *B. hermsii* are indicated on the *B. hermsii* DAH sequence, and black lines show where those regions are in each genome (23). The colored boxes shown across each isolate's plasmid indicate areas of collinearity between isolates where similarly colored boxes correspond to each other across taxa. Histograms within these boxes correspond to the level of nucleotide similarity; the higher the bar, the higher the similarity. Colored boxes below the mid-line correspond to inversions compared to the reference sequence in this alignment (*B. anserina* BA2). Genes are shown for each isolate, below the collinearity boxes, with genes above the mid-line in the positive sense and those below in the negative sense.

Block A was variable in length across species due to the number of CRASP-related genes. The size of this block ranged from ~6 kb in *B. anserina* BA2 to ~37 kb in *B. hermsii* YOR and was the largest in *B. hermsii* DAH and YOR and *B. puertoricensis* SUM (Fig. 2). The genes located here were identified by InterProScan as encoding proteins that contained a Bbcrasp-1 domain (PF05714). On average, there were 16 of these genes per isolate. *B. anserina* BA2 had the least with six genes, while *B. hermsii* YOR had the most with 30 (Supplementary file 1). Interestingly, the number of genes was not associated with the number of nucleotide repeats. For example, block A of *B. coriaceae* Co53 and *B. puertoricensis* SUM had similar numbers of BbCRASP-1 domain containing proteins. However, block A of *B. coriaceae* Co53 contained fewer repetitive sequences compared to *B. puertoricensis* SUM (Fig. 2).

Block B was evident across all species and isolates and ranged from ~14 kb in *B. anserina* BA2 to ~33 kb in *B. hermsii* YOR (Fig. 2). The proteins encoded by the genes found in block B were typically classified by InterProScan or GenBank's Prokaryotic Genome Annotation Pipeline (PGAP) as a hypothetical protein. However, InterProScan classified one BbCRASP-1 domain containing protein in *B. puertoricensis* SUM, two in *B. venezuelensis* RMA01, three in *B. parkeri* SLO, and one each in *B. turicatae* 91E135 and BTE5EL (Supplementary file 1).

Repetitive nucleotide sequences in block C were apparent in all species except *B. anserina* BA2, *B. puertoricensis* SUM, and *B. venezuelensis* RMA01 (Fig. 2). The size of block C ranged from ~1.6 kb in *B. anserina* BA2 to ~16.5 kb in *B. hermsii* DAH. The majority of genes found in this repetitive block encode proteins annotated by InterProScan as containing a domain related to Bbe16, which is also designated the borrelial persistence in ticks protein A (BptA) (14). The *bptA* gene is a single copy gene and is essential in *B. burgdorferi* (14); however, 2–14 copies of *bptA* were present in all WHsTBRF spirochete isolates (Supplementary file 2). Phylogenetic analysis of the nucleotide sequence of these

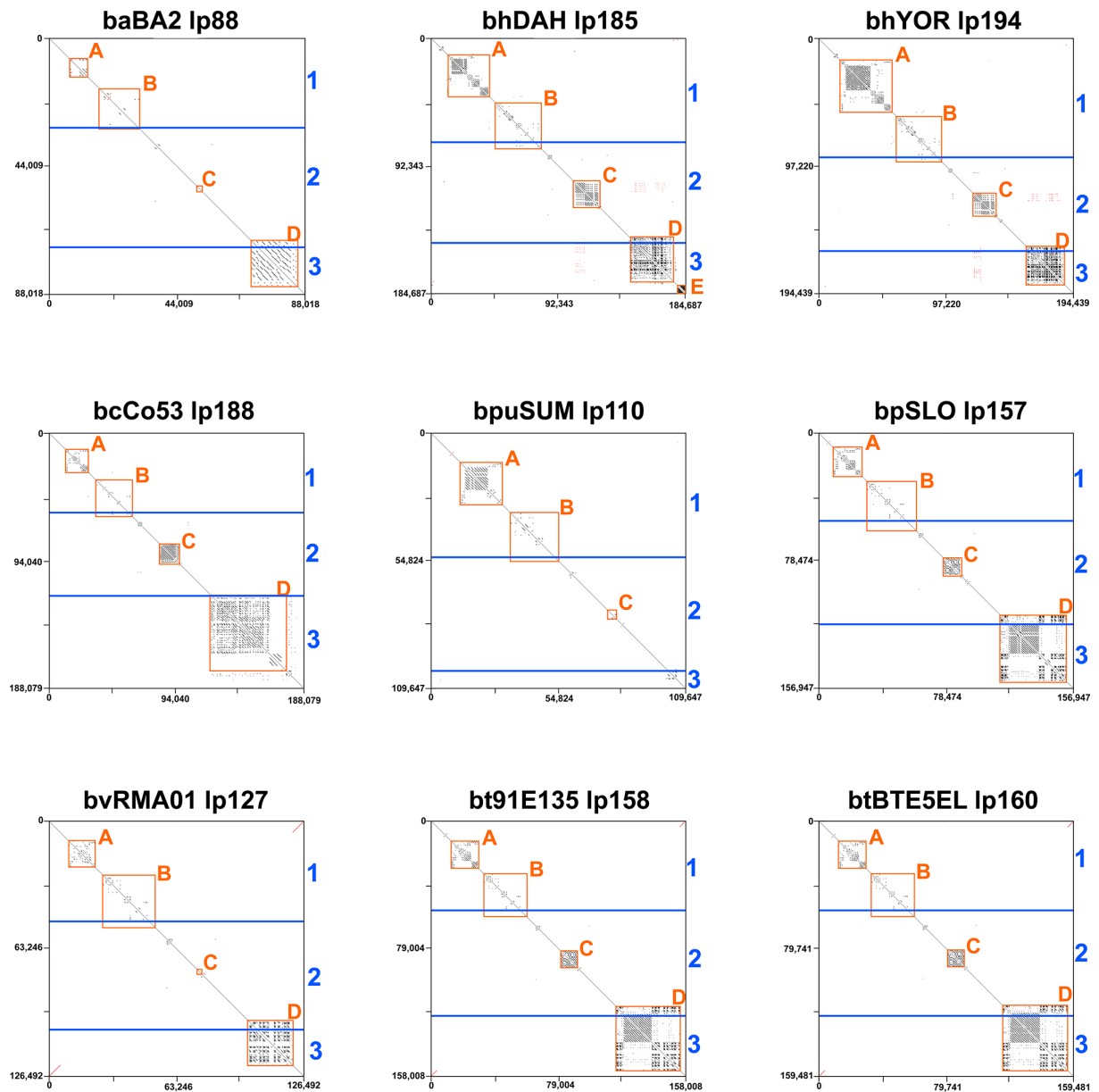

**FIG 2** Self-dot plot of the megaplasmids. Self-dot plots were generated to observe differences in the repetitive gene content across the clade. Black lines across the diagonal from top left to bottom right indicate similar sequence. Black lines deviating from the diagonal from top left to bottom right indicate areas of direct repeats. Red lines from bottom left to top right indicate areas of inverted repeats. The blue lines and numbers indicate the locations of the three conserved regions in the megaplasmids. The orange boxes and numbers indicate the locations of the five repetitive gene blocks.

putative *bptA* genes, including two genes from *B. burgdorferi* B31 (GCF_000008685.2) whose proteins were annotated with a BptA domain, identified two clades (Fig. 2; Fig. S1). One clade contained both *B. burgdorferi bptA* and a known related gene in *B. burgdorferi*, *bbj47* (27). Each WHsTBRF spirochete assembly had one gene that clustered with *bbj47*. The other clade contained all other WHsTBRF genes that contained a *bptA* domain. These genes were highly divergent from the *B. burgdorferi bptA*, with apparent duplication events occurring within each species (Fig. S1). Interestingly, *bptA*-like genes from *B. hermsii* isolates clustered to the exclusion of all other species' *bptA*-like genes. The species that were missing repetitive nucleotide sequences in block C also had the fewest number of *bptA*-like genes (Fig. S1). Moreover, some of these genes were substantially truncated (>50% of sequence missing) and are likely pseudogenes. We did not

consider these loci (bhDAH_001245, bhDAH_001246, bhDAH_001248, bhDAH_001249, and bvRMA01_000992) in our phylogenetic analysis.

Block D was the most variable block in both sequence similarity and length. This block ranged in size from non-existent in *B. puertoricensis* SUM to ~53 kb in *B. coriaceae* Co53. The amount of repetitive nucleotide sequence in block D of *B. coriaceae* Co53 was less compared to other species (e.g., *B. turicatae* and *B. parkeri*) (Fig. 2). In both *B. hermsii* isolates and in *B. coriaceae* Co53, the gene products in this block were annotated as hypothetical proteins and not classified by InterProScan (Supplementary file 1). *Borrelia anserina* BA2 had only one classified gene in block D, a phage fiber gene. *Borrelia parkeri* SLO had three genes that coded for a DUF1617 protein, a collagen-like protein, and a trimeric UDP-N-acetylglucosamine acetyltransferase (LpxA)-like enzyme family. *Borrelia venezuelensis* RMA01 had a single gene that was classified, which contained a conserved domain in the phosphatidylinositol phosphate kinase (PIPK) catalytic family. Lastly, both *B. turicatae* isolates contained only two genes classified as DUF1617.

Repetitive nucleotide block E was only present in *B. hermsii* DAH with a single gene (Fig. 2). bhDAH_001311 is a large hypothetical gene (5,513 bp) coding for an ~210-kda protein. BLASTn and BLASTp analyses of this gene failed to identify homologs outside of *Borreliaceae* (28). Within the WHsTBRF spirochetes, BLASTp found similar proteins (~40% sequence similarity, 99% query coverage, *E*-value = 0.0) in *B. puertoricensis* SUM (bpuSUM_001813, linear plasmid 46 "lp46"), *B. parkeri* SLO (bpSLO_001255, lp28), and in *B. venezuelensis* RMA01 (bvRMA01_001103, lp25). Interestingly, these plasmids are not in the same plasmid family (1). Furthermore, both BLASTn and BLASTp did not detect genes or proteins similar to bhDAH_001311 in *B. hermsii* YOR nor any other *B. hermsii* genomic group II sequences deposited in GenBank.

## F27 plasmid family analysis

The F27 plasmid family was the shortest linear plasmid family found in all WHsTBRF species (1). The size ranged from ~10 kb to ~12 kb. Previously, this family of plasmids was reported to be either a circular or a linear 5- to 6-kb plasmid (29–31). However, using long-read sequencing data and manual inspection, we hypothesized that the F27 plasmid family was linear with evidence of an inverted duplication event (Fig. S2A through C). Our rationale was threefold. First, contigs that were assembled for this plasmid family often had at least one complete and one incomplete telomeric sequence, which would be uncommon for a circular or linear plasmid (Fig. S2A through C). Second, the longest Oxford Nanopore Technology (ONT) reads of the F27 plasmids were approximately twice the size of the final plasmid sequence (Fig. S3A). These lengths suggested linearity because the telomeres of linear plasmids in *Borrelia* spp. are covalently linked (32), and both the positive and negative sense strands would be sequenced. This would cause long inverted telomeric repeats (Fig. S2A through C) (32, 33). This is opposed to circular plasmids where the longest reads could only be the true size of the plasmid. We provide examples of circular plasmid (Fig. S3B) and linear plasmid (Fig. S3C) read lengths from the original sequencing data for the genome assembly of these isolates to highlight these phenomena. Dot plot analysis of the longest reads showed patterns consistent with a linear plasmid that was fully sequenced around the telomeres. Interestingly, we noticed a "wavey" appearance in the dot plot of the contigs and reads in this plasmid family, which indicated physical DNA molecule translocation issues during sequencing (Fig. S2A, B, and D). Indeed, inverted duplication events can cause increased translocation rates in ONT sequencing (34). This occurs through the formation of secondary structures of the translocated DNA, which causes decreased sampling rates and lower base call accuracy.

Given the topology and size discrepancies between our F27 plasmid sequences to those of related plasmids from prior assemblies, we investigated the topology and size of this family. We envisioned three different scenarios for the F27 plasmids based on our previous sequencing data (1) (Fig. S3A): (1) ~10–12 kb linear plasmids, (2) ~10–12 kb circular plasmids, (3) ~20–24 kb circular plasmids. We proposed scenario 1 as the likely

case. Scenario 2 would require there to be reads that are equal to or shorter than 10–12 kb in the sequencing data initially collected for genome assembly. Given that we did have reads longer than 10–12 kb for these plasmids (Fig. S3A), the second scenario is unlikely unless there was a mixture of linear and circular F27 plasmids. Scenario 3 would be possible since there were reads >10–12 kb (Fig. S3A). To investigate which scenario was true, we used ONT to determine the sequence and length of DNA fragments and coupled this to ONT's adaptive sampling technique to enrich for F27 plasmid sequence over the background genomic DNA. We identified a restriction enzyme that would cleave on either side of each isolate's spacer sequence (the non-repetitive sequence in the middle of the inverted repeat) in the middle of the F27 plasmids. The use of this restriction enzyme with genomic DNA that was treated with or without S1 nuclease would yield a diagnostic fragmentation pattern when sequenced, allowing us to determine the correct scenario (Fig. 3A).

The S1 nuclease was included in this strategy since it can cleave the single stranded DNA (ssDNA) telomere of linear borrelial replicons (35). If we did not use the S1 nuclease, the intact telomeres would generate DNA fragments that are larger than their true size during ONT sequencing because both the positive and negative sense strands are connected and would sequence as one fragment. The presence of the telomere would also prevent the ligation of sequencing adapters to the ends of the DNA fragment preventing that DNA from being sequenced. Thus, if the genomic DNA was not treated, we would expect that in all cases (linear and circular), there would be little to no available open DNA ends for adapter ligation and that only sheared or degraded DNA would be sequenced (Fig. 3A, untreated). Fragments containing a telomere would be sensitive to the S1 nuclease (Fig. 3A; Fig. S1); therefore, when S1 nuclease is used, a linear plasmid should have sequencing reads present at or below its true size, provided both telomeres are cleaved. Circular plasmids should be relatively insensitive to S1 nuclease, although we do recognize that the ssDNA in hairpin heads could be cleaved by S1 nuclease as well (36).

When the restriction enzyme was used alone, we anticipated observing differences in sequencing fragments between the three scenarios (Fig. 3A, RE). For example, in scenario 1 (linear 10-kb plasmid) of Fig. 3A, we expected fragment B to sequence efficiently since both DNA ends are open for adapter ligation. The presence of fragment B would also confirm the inverted repeat nature of the F27 plasmids since the enzyme selected would cut on either side of the spacer sequence in the middle of the plasmid. However, when only using the restriction enzyme, the A fragments would contain the telomere and would sequence as twice the size, yielding fragment C in Fig. 3A. Fragment C would also have a lower sequencing efficiency since only one end of the DNA fragment would have a sequencing adapter. In scenarios 2 and 3, the restriction enzyme-only treatment would yield fragments B and C, which would have both DNA ends open for adapter ligation and would sequence to their true size.

We also expected diagnostic fragmentation patterns when both the restriction enzyme and S1 nuclease digestions were performed on the same genomic DNA (Fig. 3A, S1 + RE). In scenario 1, fragment C would be sensitive to the S1 nuclease and would sequence to approximately half its size, generating fragment A (Fig. 3A). In scenarios 2 and 3, the C fragment would be insensitive to the S1 nuclease, and the results would look similar between the restriction enzyme-only-treated samples and the S1 and restriction enzyme-digested samples. The diagnostic difference between scenarios 2 and 3 is the presence of reads >10–12 kb in the S1 nuclease-treated samples, and in the initial sequencing data, we collected for genome assembly that was randomly fragmented with a transposase (Fig. S3A).

Our digestion and sequencing strategy indicated the presence of A and B fragments in all isolates for the restriction enzyme and S1 nuclease double digestions, supporting our initial hypothesis (scenario 1) of ~10–12 kb linear plasmids (Fig. 3B). However, the presence of fragment C in some double digestions complicated our ability to confirm the sole presence of one of the three scenarios. Five isolates (*B. hermsii* DAH, *B. puertoricensis*

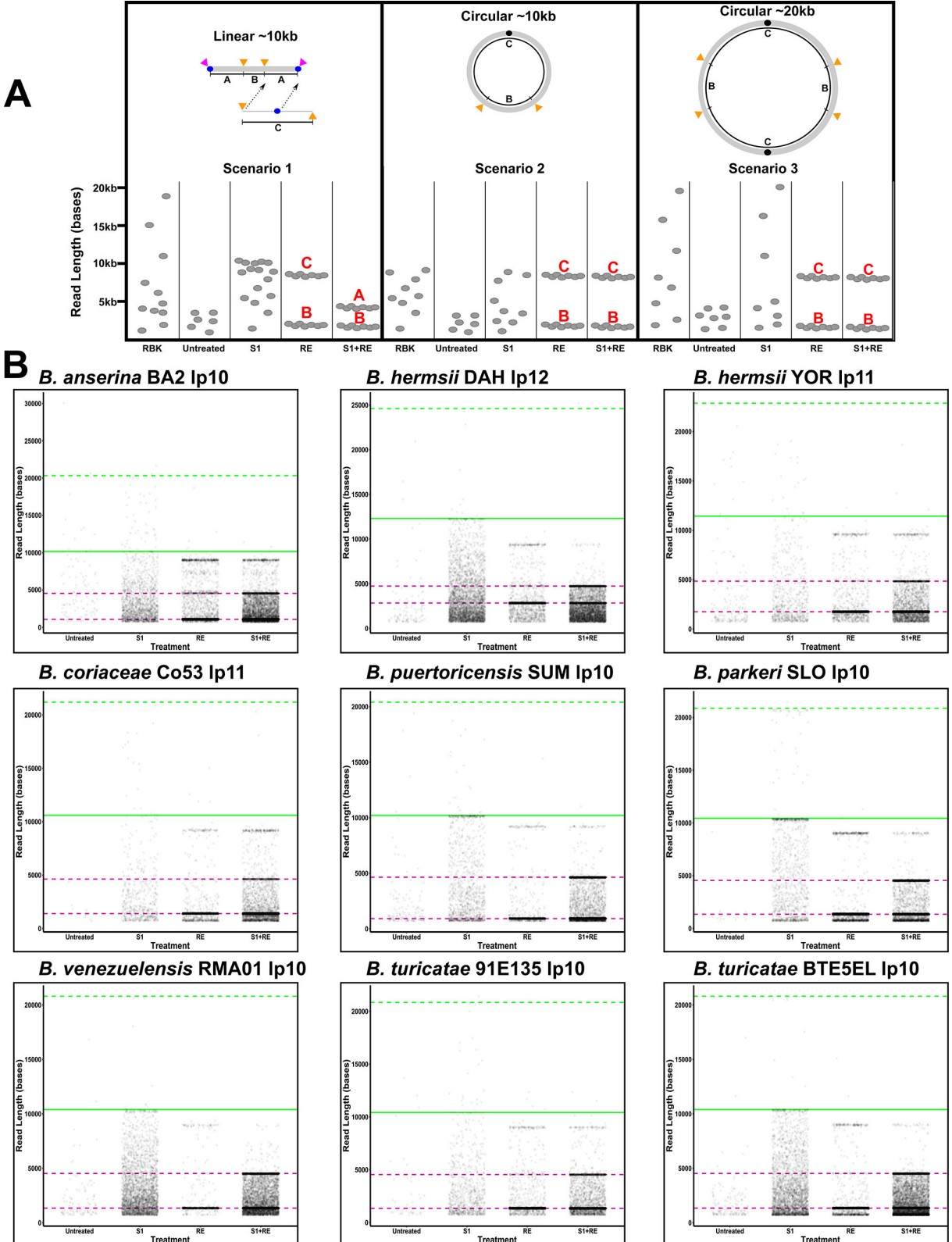

**FIG 3** Enzymatic digestion and adaptive sampling of the F27 plasmids. The three envisioned scenarios for the F27 plasmids are shown (A). In the left panel, the orange and pink triangles indicate where the restriction enzyme and the S1 nuclease would cleave, respectively. The telomeres are represented by the blue circles in the linear plasmid scenario (left panel). In the middle and right panels, the black circles on the circular plasmids indicate where the telomere would be (Continued on next page)

**FIG 3** (Continued)

relative to the linear plasmid. Below the plasmids are the predicted read lengths of the F27 plasmid for each condition. RBK indicates the predicted results for each scenario from the initial random fragmentation read data set used for genome assembly (see also Fig. S3A). Untreated represents genomic DNA that was not treated with enzymes. S1 indicates genomic DNA that was digested with S1 nuclease. RE is genomic DNA that was digested with a restriction enzyme. S1 + RE is genomic DNA that was cleaved first with S1 nuclease and then the restriction enzyme. Expected fragment sizes are labeled with a red letter and correspond to the fragments shown in the drawings. The read lengths of the primary reads that mapped to the F27 plasmid for each isolate are shown (B), with graphs following the same labeling scheme as shown in (A). The dotted purple lines indicate where the A and B fragments are predicted for each isolate. The solid green line indicates the assembled length for that isolate's F27 plasmid, whereas the dotted green line indicates twice the length of the assembled plasmid.

SUM, *B. parkeri* SLO, *B. venezuelensis* RMA01, and *B. turicatae* BTE5EL) were relatively unambiguous in that there was clear demarcation in the S1 reactions (Fig. 3B; Fig. S1) at the assembly size for the F27 plasmid with little residual fragment C in the double digestions (Fig. 3B, S1 + RE). This indicated that the telomeres were cleaved, and the adapters were able to ligate to both ends of the plasmid for full-length sequencing. The four remaining isolates (*B. anserina* BA2, *B. hermsii* YOR, *B. coriaceae* Co53, and *B. turicatae* 91E135) did not have a defined clustering of reads at the assembly size in the S1 reactions (Fig. 3B; Fig. S1) and had a greater amount of fragment C in the S1 and restriction enzyme double digestions (Fig. 3B, S1 + RE). This result suggests a mixture of topologies and sizes for the F27 plasmids. However, the presence of the C fragment in the double enzyme digestions could also be due to incomplete cleavage of the telomere by the S1 nuclease.

We further evaluated the findings by mapping and quantifying fragment abundance. Mapping the reads for each reaction condition and visualizing the read mappings using the Integrative Genomics Viewer (IGV) confirmed that the fragment sizes corresponded with their predicted sequence (Fig. S4 to S12). Quantifying fragment amounts in the restriction enzyme and S1 double digests found that fragment C was always in the minority. There was an average of 44 times more fragment A versus fragment C (although this ranged widely from 2.4× to 183×) (Fig. S13). Given this, the plasmid in scenario 1 (~10–12 kb linear plasmid) is the dominant form compared to the other scenarios considered.

Alignment of the F27 plasmid family showed that the genes found on these plasmids were generally conserved across species (Fig. 4). Interestingly, the plasmid-partitioning genes in this plasmid family were in a unique configuration compared to other plasmid families, and they lacked PF50 genes (1). Aside from the plasmid partitioning genes, there were five to seven other predicted genes on the F27 plasmids that were annotated to code for hypothetical proteins. The spacer sequence between the inverted duplication sequences was variable in length and sequence identity across species, was largely noncoding, and was not found anywhere else on the F27 plasmid or in other replicons in that plasmid's genome assembly. However, *B. hermsii* DAH had a transposase pseudogene recently annotated in this region in the RefSeq assembly (bhDAH_RS05685). We performed BLASTn analysis on the bhDAH_RS05685 locus and found nucleotide identity to *B. hermsii* YOR and *B. burgdorferi* and *Borreliella* (*Borrelia*) *mayonii* isolates. In the *B. hermsii* isolates, the BLASTn score (98%–100% coverage, *E*-value <5e−53, sequence identity between 76.5% and 100%) fell within the F27 plasmid in the spacer sequence in the same general location as in *B. hermsii* DAH. For the *B. burgdorferi* isolates, significant hits (coverage >99%, *E*-value <2e−20, and sequence identity >71%) were either in the lp25 plasmid (e.g., MM1, JD1) in genes annotated to encode either a mobile element protein, transposase-like protein, or transposase, IS605 family (BbuMM1_E180, BbuMM1_E210, BbuJD1_E18, BbuJD1_E24, BBU118A_E13, BBU118A_E17), or on the lp38 plasmid (e.g., B31_NRZ and B408) in a pseudogene that was annotated either as a transposase (B1U23_05860) or unannotated (isolate B408). For the *B. mayonii* isolates (MN14-1420 and MN14-1539), the hit was in the lp28-3 plasmid in a pseudogene annotated as a transposase (Bmayo_04865 and A7X70_05675). Collectively, our data indicated that the F27 plasmids were largely a linear ~10–12 kb plasmid formed by

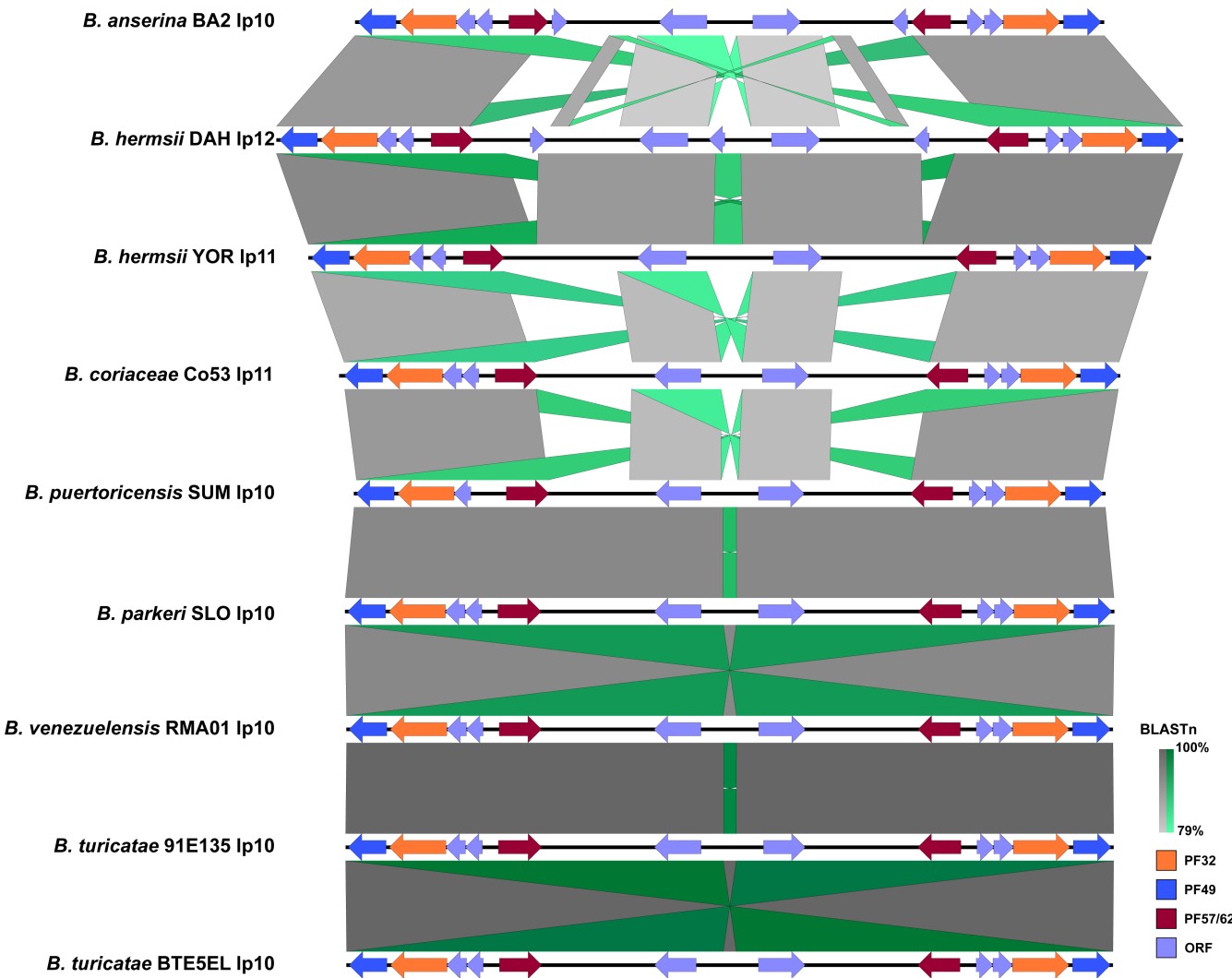

**FIG 4** Alignment and visualization of the F27 plasmid family. The minimum BLASTn alignment was set to 250 nt. BLASTn results scale from 79% to 100% going from light to dark gray. Nucleotide inversions scale similarly but from light to dark green. Open reading frames (ORFs) and plasmid partitioning loci are indicated by specific colors.

an inverted repeat with two symmetrical halves that contain a unique organization of plasmid-partitioning genes and other genes with unknown function.

## F28 plasmid family analysis and comparing antigenic variation systems in WHsTBRF spirochetes

Lyme disease- and RF-causing spirochetes employ antigenic variation systems to evade host immunity, and this has been extensively studied for RF spirochetes in *B. hermsii* (37–39). The protein family driving antigenic variation for RF spirochetes is designated the variable major proteins (Vmps) (40). Antigenic variation is achieved through the recombination of silent, archived genes into a single expression site (41, 42). Since there is only one *vmp* expression locus in a single bacterium, we refer to both the expressed and silent archived *vmp*s as "alleles" based on the nomenclature put forth by Rich et al. (43). Our classification of *vmp*s includes predicted pseudogenes as annotated by PGAP. The predicted protein products of pseudogenes were on average ~40%–50% shorter than the length of predicted Vmp alleles. Furthermore, Vmps are classified by molecular weight, with the variable small proteins (Vsps) at ~22 Kda and variable large proteins (Vlps) at ~37 Kda (44–46). The antigenic variation expression site in *B. hermsii* uses a

σ70-like promotor (47) and is characterized by upstream genetic elements that include three stem loop structures and a transcription-enhancing tract of ~12–16 thymines (42, 48, 49). In our analysis, the *B. hermsii* promoter sequence and the thymine tract were found exclusively on the *B. hermsii* F20 plasmids, which were lp26 and lp23 plasmids for DAH and YOR isolates, respectively (Fig. S14). We also identified the upstream and downstream homology sequences (UHS and DHS, respectively) on these two plasmids (Fig. S14). The UHS and DHS were previously hypothesized as the two areas of sequence homology, which allow directed recombination to occur in *B. hermsii* (42). However, analysis of the other WHsTBRF spirochete F20 plasmids indicated that this plasmid family was not syntenic, and we were unable to identify a similar *vmp* expression site in other species (Fig. S15).

Since *B. turicatae* utilizes a *B. burgdorferi ospC*-like promoter to express *vmp*s (50, 51), we analyzed the genome assemblies for the presence of this promotor to identify the expression sites. The promoter sequence was identified on the F28 plasmid family (Fig. 5A). Interestingly, for *B. hermsii*, the promoter on these plasmids regulates the expression of the variable tick protein (*vtp*) gene (discussed further below). Sequence alignments of the F28 plasmid family determined that the −35 and −10 sites and the ribosomal binding site were largely conserved, with the −10 site possessing nucleotide variability (Fig. 5A). The −10 site was conserved in *B. puertoricensis* SUM, *B. parkeri* SLO, *B. venezuelensis* RMA01, and *B. turicatae* 91E135 and BTE5EL. However, these species also had the pseudo −35 site immediately upstream of the −10 site, which was previously hypothesized to cause RNA polymerase holoenzyme mis-alignment, thereby potentially requiring a trans-activating protein (47). As previously described for *B. turicatae* (50, 52, 53), the promoter was downstream of a gene encoding an oligopeptide permease-like protein for all genome assemblies except *B. anserina* BA2, which was missing this gene.

We also attempted to identify sequence analogous to the UHS and DHS of the *B. hermsii* F28 plasmids. We detected a UHS sequence across all species (Fig. 5A). However, we did not detect a DHS sequence. The sequence downstream of the expression site, including the *vmp* loaded there, was nearly identical to other plasmids for all genome assemblies except *B. coriaceae* Co53 (Table 1). The plasmids that had sequence similar to the expression plasmid did not fall into any particular plasmid family (Table 1) nor did they contain a complete promoter upstream of the *vmp* found in the F28 *vmp* expression site. In *B. coriaceae* Co53, we found no sequence in the region where the DHS would be that had sequence identity >90% to other parts of its genome. We were also not able to find the archived gene of the *vlp* in the *B. coriaceae* Co53's putative *vmp* expression site.

There were additional structural differences in the *vmp* expression locus between species of RF spirochete. Similar to what was reported for *B. turicatae* (50, 52, 53), the putative *vmp* expression site located on the F28 plasmids was internal with *vmp* alleles and non-*vmp* genes downstream (Fig. 5B). This is contrary to the *B. hermsii vmp* expression site on the F20 plasmids, which is terminal on the telomere and only one *vmp* is present and downstream genes are absent (Fig. S14) (42).

Intriguingly, analysis of the F28 family showed that *B. venezuelensis* RMA01 had two nearly identical plasmids in the F28 family, lp35 and lp37. Both *B. venezuelensis* RMA01 F28 plasmids contained the o*spC*-like expression site. However, lp37 had a pseudogenized *vsp*, caused by a point mutation resulting in a frameshift in this gene, whereas lp35 had an intact *vsp* gene. The *vmp* alleles located downstream of the expression site for lp35 and lp37 are different in number and sequence, and both plasmids differ at the 3′ ends. The 3′ end of lp35 matches the 5′ telomeres of lp30 and lp31. The 3′ end of lp37 matches the 3′ end of lp30 and 5′ telomere of lp25. These data suggest that a recombination event occurred that replaced the entirety of the 3′ end rather than just *vmp* loci downstream of the expression site.

We confirmed the presence of the two 3′ end configurations for *B. venezuelensis* RMA01 F28 plasmids. Mapping of long reads that were initially used for the assembly of the *B. venezuelensis* RMA01 genome identified reads specific to both 3′ end configurations (Fig. S16). Adaptive sampling using the ONT MinION platform was also performed

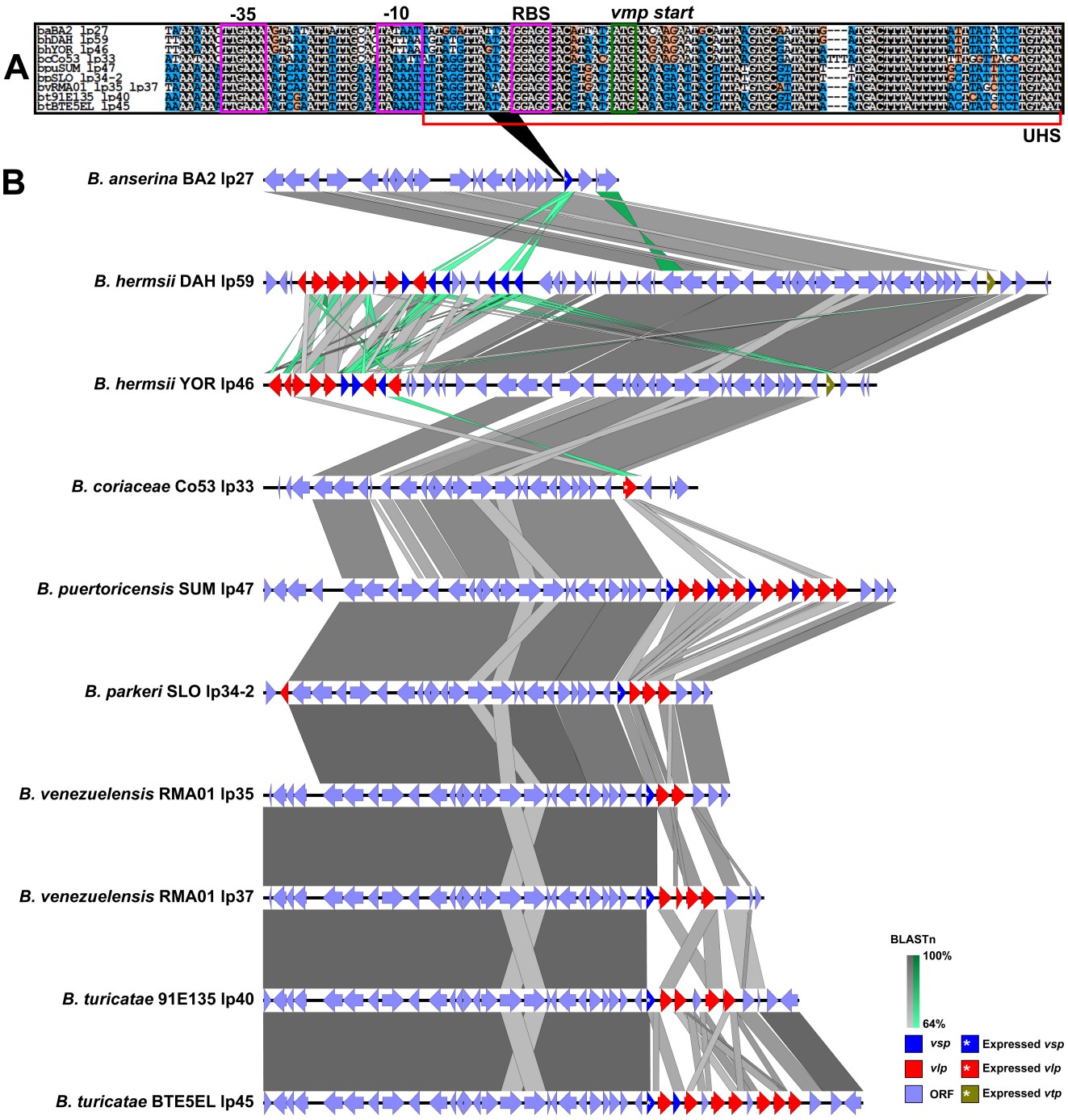

**FIG 5** Alignment and visualization of F28 plasmid family and *vmp* promoter. (A) The *vmp* promoter alignment is seen in the top sequence. Promoter features are annotated and boxed in purple (RBS, ribosome binding site). The ATG start codon is boxed in green. The upstream homology sequence (UHS) is indicated with a red bracket. For the alignment, a white text in a black square means the sequences are the same, black text in a white square indicates sequences are different, black text in an orange box indicates a transition substitution, and white text in blue squares indicates that that nucleotide matches the consensus for that position. (B) The plasmids were aligned using EasyFig. This analysis demonstrated sequence conservation and synteny across species. The minimum BLASTn alignment was set to 250 nt. BLASTn results scale from 64% to 100% going from light to dark gray, whereas inversions scale the same but from light to dark green. Open reading frames (ORFs) and the *vmp* alleles are shown in specific colors. The *vmp* in the expression site is indicated by an asterisk within the indicated ORF.

**TABLE 1** Plasmids with similar sequence downstream of the F28 expression site

| Genome | F28 plasmid | Plasmid with similar sequence | BLASTn % sequence coverage | BLASTn % sequence identity | Plasmid family |
|---|---|---|---|---|---|
| *B. coriaceae* Co53 | lp33 | None | N/A[a] | N/A | N/A |
| *B. puertoricensis* SUM | lp47 | lp39 | 96 | 99.97 | F16 |
| | | lp46 | 100 | 99.96 | F14 |
| | | lp57 | 98 | 99.37 | F18 |
| | | lp65 | 100 | 99.98 | F1 |
| *B. parkeri* SLO | lp34-2 | lp34-1 | 99 | 99.97 | F7 |
| | | lp60 | 99 | 99.96 | F29 |
| *B. venezuelensis* RMA01 | lp35 | lp30 | 100 | 100 | F12 |
| | | lp31 | 100 | 97.13 | F10 |
| | lp37 | lp25 | 97 | 99.82 | F14 |
| | | lp30 | 99 | 99.97 | F12 |
| *B. turicatae* 91E135 | lp40 | lp38 | 100 | 99.96 | F19 |
| *B. turicatae* BTE5EL | lp45 | lp28 | 99 | 99.94 | F14 |

[a]N/A, not applicable.

to selectively sequence the lp35 and lp37 plasmids of *B. venezuelensis* RMA01 and the related lp40 plasmid of *B. turicatae* 91E135. Adaptive sampling data supported the two different 3′ end configurations of *B. venezuelensis* RMA01's lp35 (Fig. S17) and lp37 (Fig. S18) plasmids, while only one 3′ end configuration was present in *B. turicatae* 91E135's lp40 plasmid (Fig. S19). Taken together, these data confirmed that *B. venezuelensis* RMA01 contained two nearly identical plasmids with the *vmp* expression locus but differed in their 3′ ends.

We further investigated the antigenic variation systems of the WHsTBRF spirochetes by determining the number of *vmp* alleles for each isolate. The number varied among species, but *vlp* alleles outnumbered *vsp* alleles in every genome assembly (Fig. S20). Surprisingly, *B. anserina* BA2 contained only one *vsp* found on the F28 expression plasmid and no *vlp*s. A phylogenetic analysis of *vmp* alleles determined relatedness within *vsp* and *vlp* groups. We did not observe phylogenetic structure in *vsp* alleles, which agreed with what Kuleshov et al. (29) reported for *B. miyamotoi vsp* alleles (Fig. S21A). Phylogenetic analysis of the *vlp* alleles separated them into the four previously described subfamilies (alpha, beta, gamma, and delta) (54). The exception was *B. anserina* BA2, which did not have *vlp* alleles (Fig. S21B). We analyzed each isolate individually to determine the subfamily designation for each *vlp* allele (Fig. 6; Supplemental File 4). The proportions of each *vlp* subfamily indicated that the beta subfamily *vlp* alleles remained relatively constant, while the alpha, gamma, and delta subfamilies varied between isolates (Fig. S22).

## Assessment of the *vtp* expression site between species

The F28 plasmids in *B. hermsii* contained the *vtp* expression site (55, 56), and we performed an analysis to identify orthologs in other species of WHsTBRF spirochetes. The Vtp is a Vsp with a unique signal peptide sequence (55), exists in only one copy in the *B. hermsii* genome (47), and is controlled by a different promoter compared to the antigenic variation system (57). The *vtp* is expressed during spirochete colonization of the salivary glands (56) and during early mammalian infection (58), and is crucial for infecting the host (17). We assessed our *vsp* phylogeny to see if we could detect a difference in the phylogenetic relationship between *vtp* genes (bhDAH_001490 and bhYOR_001299) and other *vsp* alleles (Fig. S21A). There was no discernable difference. This was not surprising since the only major difference between Vtps and Vsps is the signal peptide sequence (44, 55). Consequently, we evaluated the presence of Vtp in the WHsTBRF spirochetes in our data set through cluster analysis of signal peptides (first ~20 amino acids) of Vsps. Only the *B. hermsii* Vtp, *B. burgdorferi* OspC [previously shown to have a signal peptide similar to Vtp (55)], and *B. anserina* BA2 Vsp signal peptides clustered together indicating

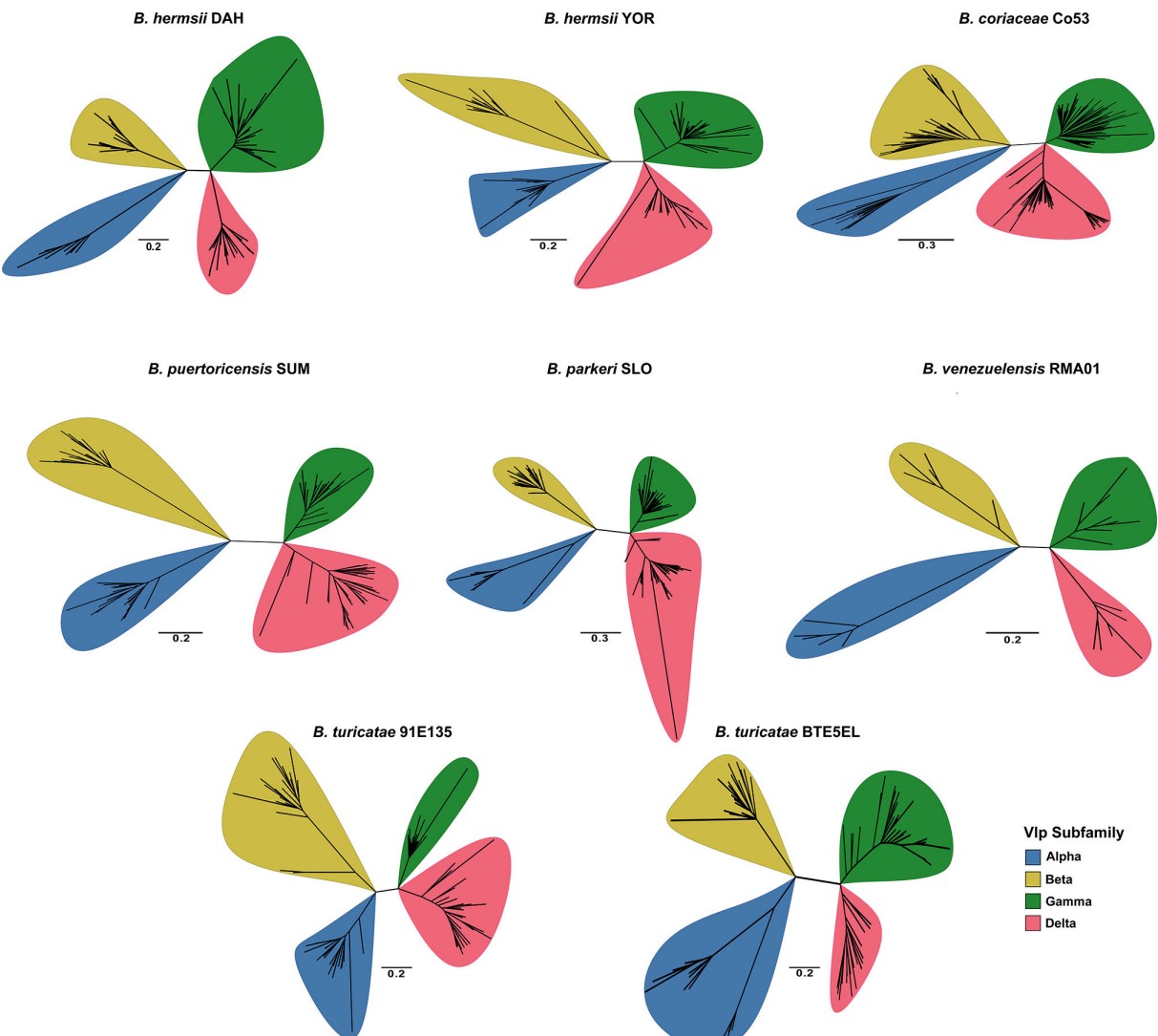

**FIG 6** Phylogenetic analysis of the *vlp* alleles of individual WHsTBRF spirochete isolates. Each isolate's *vlp* allele nucleotide sequences were used to infer a maximum likelihood tree with 1,000 bootstrap replicates. Branches with less than 50% support were collapsed. The scale bar indicates substitutions per site. Note, *B. anserina* BA2 did not have any *vlp* alleles, so it is not shown here.

that there were no other Vtp-like signal peptides that had >80% amino acid sequence identity. The Vsp from *B. anserina* BA2 had 90% similarity to the OspC signal peptide. This similarity between the OspC and the *B. anserina* Vsp signal peptides was consistent to findings reported by Schwan et al. (15).

## DISCUSSION

We investigated the F6, F27, and F28 plasmid families because they were conserved in all species of WHsTBRF spirochetes with genome assemblies currently available. The F6 family (megaplasmid) is the largest linear plasmid reported for RF spirochetes, while the F27 plasmid family is the smallest plasmid (~10 kb) and had no close relative in the LD spirochetes (1). We also characterized the F28 plasmid family, which contained the *vtp* gene of *B. hermsii*; however, in the other species, these plasmids contained the expression locus for antigenic variation. Collectively, our findings identified distinctions in the repetitive gene content and antigenic variation systems across the species of WHsTBRF spirochetes.

In RF spirochetes, the F6 megaplasmid is hypothesized to be important for tick colonization and mammalian infection based on its gene content and transcriptional profile during vector colonization and mammalian host infection (23, 59, 60). For example, a gene expression analysis of the *B. turicatae* megaplasmid indicated that ~67% of the genes found on the megaplasmid were up-regulated when grown under conditions mimicking the tick environment (59). Further analysis validated the up-regulation of genes localized at the 3′ end of the megaplasmid in the tick compared to infected mouse blood. In addition, the megaplasmid is related to the essential lp54 plasmid in LD-causing spirochetes (4, 22). Lp54 contains genes encoding outer surface proteins, such as outer surface lipoproteins A and B, Csp*A*, and decorin binding proteins A and B, which play important roles in tick colonization and vertebrate infection (61–65).

Previous work with the megaplasmid identified three regions of conservation with five blocks of repetitive nucleotide content in *B. hermsii*, *B. turicatae,* and *Borrelia duttonii* (23). Our work supported these findings for the additional species of WHsTBRF spirochete evaluated. However, the length of the three conserved regions varied across the species, which was driven by the number of repetitive sequences in the five blocks. Two repetitive areas that stood out were blocks A and C. For example, block A of the megaplasmid contains homologs of *B. burgdorferi* CRASP genes. In *B. burgdorferi*, the proteins are important for complement resistance in the vertebrate host (66–69). Interestingly, these genes were variable in number and repetitive sequence across the WHsTBRF spirochetes. *Ornithodoros* species are known to feed on a variety of vertebrate hosts (70–75), and the observed variability in CRASP homologs may contribute to complement resistance traits that result in the diversification of susceptible hosts.

Repetitive region C was variable across isolates and contained multiple copies of *bptA* homologs. BptA is a surface exposed lipoprotein that is essential for persistence of *B. burgdorferi* in *Ixodes scapularis* (14). Multiple copies of *bptA*-like genes were found in prior work on the megaplasmids of *B. hermsii* and *B. turicatae*; however, their functional role remains uninvestigated (23). A microarray analysis of the megaplasmid in *B. turicatae* reported over a threefold up-regulation of expression of these genes in tick-like *in vitro* growth conditions relative to spirochetes isolated from mammalian blood (59). Transcriptional profiles and the gene's duplication and diversification in WHsTBRF spirochetes suggest that they may be important in vector colonization. Indeed, gene duplication and diversification have been linked to environmental adaption in other prokaryotes (76, 77). The megaplasmid in RF spirochetes warrants further investigation to determine its role in the infectious life cycle of these pathogens.

Our analysis also indicated that the F27 family is conserved in species of WHsTBRF spirochetes. In prior RF spirochete genome assemblies, F27 plasmids were often reported as a 5- to 6-kb circular plasmid (*B. turicatae* 91E135 GCA_000012085.2 , *B. hermsii* HS1 GCA_001660005.1, and *B. anserina* Es GCA_001936255.1). However, a plasmid analysis of WHsTBRF spirochetes using pulsed-field gel electrophoresis indicated that every genome possessed an approximately 10- to 12-kb plasmid, and a phylogenetic analysis determined that a plasmid of similar size clustered into the F27 family (1). Newer third-generation sequencing technology and assembly strategies demonstrated that the F27 plasmid family was not only linear but also has undergone an inverted duplication event. Our work supported this notion using an enzymatic digestion and sequencing approach. However, the presence of a mixed population of circular and linear plasmids could not be ruled out. We speculate that our data may be confounded by chemical contaminants in the genomic DNA resulting in suboptimal reaction conditions, which may have decreased the S1 nuclease enzyme efficiency. If the S1 nuclease failed to cleave both a linear plasmid's telomeres, the reads would look like fragment C instead of fragment A. Further work is needed to tease apart if indeed a mixed population of plasmids exists. While the F27 plasmid is conserved across species, the genes are largely of unknown function, and the overall role of the plasmid remains elusive.

The presence of the transposase pseudogene in the spacer region of *B. hermsii* DAH's F27 plasmid was intriguing. This may explain how this plasmid family was generated.

These plasmids may have arisen from a mobile genetic element causing a fusion of two copies of the same plasmid. The reason the other isolates do not have sequence similarity in this region may be that these genes or pseudogenes degraded post-speciation. While this transposase pseudogene had BLASTn hits to plasmids in *B. burgdorferi* and *B. mayonii*, they were in plasmids that are not related to the F27 plasmids by PF32 or PF57/62 phylogenetic analyses (1). Two of the *B. burgdorferi* genes that matched by BLASTn analysis (BBU118A_E13 and BBU118A_E17) were annotated as a transposase, IS605 family protein. Further investigation into the role transposases and other mobile genetic elements play in driving the plasmid diversity and their impact on the borrelial spirochete life cycle is warranted.

The F28 plasmid family contains housekeeping genes and the putative expression site for antigenic variation for all species of the WHsTBRF clade except for *B. hermsii* and *B. anserina* (6, 78–81). Interestingly, compared to the terminal *vmp* expression site on *B. hermsii* F20 plasmids, the *vmp* expression sites on the F28 plasmids were internal, as previously reported for *B. turicatae* (53). Downstream of the F28 plasmid expression site were multiple *vmp* alleles and other non-*vmp* genes. Dai et al. (42) previously reported that multiple *vmp*s can be present downstream of the F20 plasmid family's antigenic variation expression site in *B. hermsii* but that further recombination events take place to remove these alleles leaving only one *vmp*. The *vmp* gene in the expression site is directly adjacent to the telomere of the F20 plasmid. Our work determined that, for non-*B. hermsii* species, there were multiple *vmp* alleles downstream of the expression site as well as non-*vmp* genes. In addition, there were long stretches of sequence downstream of the expression site, which are nearly identical to other plasmids storing the archived *vmp*. These findings were noted previously in *B. turicatae* (50, 52, 53), and our findings further highlight the potentially exceptional nature of antigenic variation system in *B. hermsii*.

Surprisingly, we found that *B. venezuelensis* RMA01 had two F28 family plasmids. These plasmids (lp35 and lp37) were identical except for the sequence from the *vmp* located in the expression site to the end of 3′ telomere. Indeed, Pennington et al. (53) reported in separate *B. turicatae* clones similar findings of extensive sequence similarity downstream of the *vmp* in the expression site between the expression plasmid and plasmids harboring the archived *vmp* gene. *B. venezuelensis* RMA01's lp35 and lp37 plasmids may exist in different populations within the polyclonal population sequenced and require further investigation via sequencing of clonally derived isolates. Future efforts comparing the evolution of the antigenic variation systems across all the RF spirochetes could identify conserved themes in the biology of these pathogens.

The *B. hermsii* F28 plasmids contain the expression site of the *vtp* gene rather than the *vmp* expression site (47); however, our attempt to identify a related *vtp* gene in the other isolates was largely unsuccessful. Our efforts to determine other putative *vtp* genes were based on the signal peptide of the protein, which is the only way to discriminate a Vsp from a Vtp (55). In agreement with previous reports (15), our findings indicated that the signal peptide of *B. anserina* BA2 was highly similar to the *B. hermsii* Vtp signal peptide (90%). The absence of a readily identifiable *vtp* gene in RF *Borrelia* spp. outside of *B. hermsii* continues to be perplexing, given its importance in the species' ability to complete the tick-mammalian life cycle (17).

Our findings of a single *vmp* in the F28 expression plasmid of *B. anserina* were noteworthy. *Borrelia anserina* animal hosts are reportedly limited to fowl where it presents with prolonged spirochetemia of ~10–14 days that continues without relapses until the infection is resolved or the bird dies (82, 83). The lack of other *vmp* alleles in the *B. anserina* BA2 genome agrees with the clinical presentation of *B. anserina,* which suggests that this species is incapable of antigenic variation. This finding is interesting because other RF species in the hard and soft tick clades have numerous *vmp* alleles necessary for antigenic variation (29, 42, 84–86). The loss of *vmp* alleles and the inability for antigenic variation in *B. anserina* coupled with its comparatively small genome size could be due to niche adaptation to fowl. Consequently, the requirement to evade

the immune systems of a diverse host range is no longer necessary. Furthermore, the similarity of the *B. anserina* BA2 signal peptide for its sole Vsp to the Vtp signal peptide of *B. hermsii* was also intriguing. However, unlike *B. hermsii* Vtp, the Vsp of *B. anserina* is expressed throughout infection in the avian host (87). Given what has been reported previously for *B. anserina* and in light of our data and the clinical manifestations of this pathogen, the single Vsp for *B. anserina* BA2 may have a more nuanced function compared to conventional Vsp and Vtp proteins (82, 83, 87–89).

Our analysis is currently limited by the number of isolates available and the functional annotation of many borrelial proteins. While the more heavily studied species (*B. hermsii*, *B. turicatae*, and *B. parkeri*) have multiple isolates available, others have only one isolate (*B. puertoricensis* SUM and *B. venezuelensis* RMA01) or there are limited numbers of isolates (*B. anserina* and *B. coriaceae*). As more isolates are analyzed, a refined understanding of intraspecies plasmid conservation will be accomplished. Perhaps one of the biggest limitations is that functional annotation of borrelial proteins is poor. A considerable number of genes were annotated as "hypothetical proteins" or "domain of unknown function containing proteins," which complicates comparative genomics analysis. Moreover, we built on prior work with *B. turicatae* and its promoter to identify the putative expression locus for the *vmps* in the remaining species evaluated. We did not check every *vmp* allele in this data set (~1,000 predicted *vmp* alleles) for a complete promoter and ribosome binding site; there could be novel expression sites from what we investigated. Transcriptional studies are needed to determine if the F28 plasmid houses the expression locus for antigenic variation in species other than *B. turicatae*. These shortcomings represent opportunities for future work to understand the diversity and genome biology of RF spirochetes and how they relate to human health.

Given the unique complexity of borrelial genomes and our limited understanding of the role of plasmids in RF spirochete biology, assessment of plasmid-resolved genome assemblies is important. This current study sets the foundation to evaluate the role of conserved plasmids in pathogenesis and vector competence. Given the advances in TBRF spirochete genetics (17, 90–92), studies can now be performed to assess the role of these plasmids and the genes they contain in the life cycle of RF spirochetes.

## MATERIALS AND METHODS

### Genome assemblies

The genome assemblies used in this work were generated by our laboratory previously (1). The GenBank accession for these assemblies were as follows: *B. anserina* BA2 (GCA_023035575.1), *B. hermsii* DAH (GCA_023035675.1), *B. hermsii* YOR (GCA_023035795.1), *B. coriaceae* Co53 (GCA_023035295.1), *B. puertoricensis* SUM (GCA_023035875.1), *B. parkeri* SLO (GCA_023035815.1), *B. venezuelensis* RMA01 (GCA_023035835.1), *B. turicatae* 91E135 (GCA_023035855.1), and *B. turicatae* BTE5EL (GCA_023035415.1).

### Bioinformatics

Example commands used with the respective software are found in Supplemental File 3.

### *Dot plot analysis*

Self-dot plots of the F6 plasmid family were generated using the nucleotide sequences of each megaplasmid with the LAST aligner (93). The dot plots were annotated in Inkscape (94).

Dot plots of the F27 contigs and read for *B. hermsii* DAH were performed using FlexiDot (v1.06) (95) on the initial assembly and read data generated by Kneubehl et al. (1). The dot plots were annotated in Inkscape.

## Mauve

ProgressiveMauve (v20150226 build 10) was used to visualize sequence similarities for the megaplasmids (96). ProgressiveMauve was run without assuming collinearity with default options. Visualization was done using the Mauve program and annotated in Inkscape.

## bptA phylogenetic analysis

Using the InterProScan analysis from Kneubehl et al. (1), we identified in each isolate and in the *B. burgdorferi* B31 genome the hits for BptA using the Pfam designation PF17044 (InterProScan results for loci are located in Supplemental File 2). Nucleotide sequences were aligned using MAFFT (v7.486) using the --auto option (97). Using these alignments, we inferred a maximum likelihood tree using IQ-TREE2 with the -m MFP and -B 1000 options (98–100). The tree was visualized using iTOL (v6) and annotated in Inkscape (101).

## EasyFig

To visualize whole plasmid similarity for the F6, F20, F27, and F28 plasmid families, we used EasyFig (v2.2.5) (102). Each plasmid's GBK file was uploaded to EasyFig which was run using a minimum cutoff length of 250 nt for the BLASTn results (103). Direct sequence similarity was shown going from gray to darker gray, inverted sequence similarity was shown going from lighter green to darker green. Individual gene features that were highlighted, were done by appending the specific gene's information in the GBK files of each plasmid with "/colour=" followed by an RGB code indicating a specific color.

## Promoter analysis

Vmp expression site promoters were investigated based on previous reports in *B. hermsii* (48) and *B. turicatae* (52). The identified promoters were aligned using MAFFT and visualized using pyBoxshade.py (104). Promoter features were based on previous reports in *B. hermsii* (42, 47–49) and *B. turicatae* (50, 52, 53). These were annotated on the alignment image with Inkscape.

## vmp analysis

*Vmp* alleles were assessed using the results of each genome's InterProScan analysis from Kneubehl et al. (1) and PGAP's annot.gff file (*vsp* = PF01441, *vlp* = PF00921) per Kuleshov et al. (29) (InterProScan results for loci are indicated in Supplemental File 4). This analysis was performed using our WHsTBRF spirochete data set (1). The phylogenetic analysis for the total *vsp* and *vlp* loci tree inferences as well as individual isolate *vlp* loci trees was performed similar to the *bptA*. This same analysis was carried out for the individual isolate's *vlp* complements as well. The trees were visualized using iTOL (v6) and annotated in Inkscape (101). The total *vsp* and *vlp* alleles for each genome were graphed using Graphpad Prism 8. The *vlp* subfamilies were determined using the previously typed *vlp*s in *B. hermsii* (54). We identified similar alleles in our *B. hermsii* DAH genome using BLASTn, and these were used to type the other isolates' *vlp* alleles (103). Vtp proteins were investigated by using CD-HIT (v4.8.1) clustering of the signal peptide (first 20 amino acids) of each Vsp protein and OspC of *B. burgdorferi* B31 using a sequence identity threshold of 80% (105).

## Borrelia venezuelensis RMA01 F28 plasmid 3′telomere long-read mapping

To map long reads specific to the F28 plasmids in *B. venezuelensis* RMA01, we filtered the reads originally used to generate the *B. venezuelensis* RMA01 genome (SRR15006050) using NanoFilt (v2.8.0) with the -q 10 and -l 15,000 (106). Minimap2 (v2.24-r1122) was used to map the filtered reads to the *B. venezuelensis* RMA01 genome assembly

(GCA_023035835.1) (107, 108). Samtools (v1.11) was used to extract primary reads mapping for lp35 and lp37 using the -b, -F 0×104, and -q 30 options (109). The resulting BAM file for each plasmid was visualized in the Integrated Genomics Viewer (IGV, v2.6.2) and annotated in Inkscape. In the IGV visualizations, indels <10 bp were masked, supplementary reads were linked, and reads were sorted by read order.

## Adaptive sampling sequencing and analysis

To determine the topology and size of the F27 plasmids, we performed S1 nuclease and restriction enzyme digestion of genomic DNA coupled with adaptive ONT sampling. The same genomic DNA samples used to generate the genome assemblies of each of the isolates (1) were used for F27 plasmid sequencing. The S1 nuclease (Thermo Scientific) was used to cleave the ssDNA telomere sequences at the ends of linear plasmids. The S1 nuclease reaction was performed per manufacturer's instructions scaled to 60 µL with 1 µg of genomic DNA. The S1 reactions were carried out at room temperature for 1 hour. The reactions were not stopped with EDTA or heat but were purified with an equivalent volume of MagBind Total Pure NGS purification beads. The beads were incubated with the reaction at room temperature for 10 min on an end-over-end mixer, immobilized on a magnetic rack, washed twice with 200 µL of 80% ethanol, and eluted in 18 µL of Qiagen's EB buffer. The purified reaction was quantified with a Qubit 4.0 using the 1× dsDNA High Sensitivity kit. For the S1 and restriction enzyme digestions, 300 ng of purified S1 reaction was taken forward. For the restriction enzyme-only reactions, 300 ng of genomic DNA was used. The restriction enzyme was selected that would cut once on either side of the spacer sequence in the middle of the plasmid and yield diagnostic fragments to discriminate between different topology and size scenarios. The selected restriction enzymes were *B. anserina* BA2 (BglII), *B. hermsii* DAH (EcoRI-HF), *B. hermsii* YOR (EcoRV), *B. coriaceae* Co53 (EcoRV), *B. puertoricensis* SUM (EcoRV), *B. parkeri* SLO (EcoRV), *B. venezuelensis* RMA01 (EcoRV), *B. turicatae* 91E35, and BTE5EL (EcoRV). Restriction enzyme digestions were carried out following manufacturer's (New England Biolabs) instructions for a 50-µL reaction and 1 µL of restriction enzyme. The restriction enzyme reaction was carried out at 37°C for 30 min before bead purification with MagBind Total Pure NGS purification beads, as performed previously for the S1 nuclease reaction cleanup. Barcoded sequencing libraries were prepared using ONT's SQK-NBD114.24 library preparation kit. Two hundred and fifty nanograms of each condition (untreated genomic DNA, S1 nuclease-treated DNA, restriction enzyme-treated DNA, and S1 nuclease- and restriction enzyme-treated DNA for each isolate) were end-prepped and dA-tailed with the NEBNext Companion Module for ONT Ligation Sequencing following ONT's kit instructions. Barcodes were ligated to the DNA in each sample as per ONT's instructions, and 100 ng of each condition was pooled together for cleanup and adapter ligation. A total of 200 ng of barcoded and adapter-ligated DNA was loaded onto an R10.4.1 flow cell on an Mk1B sequencer. The DNA was sequenced using MinKNOW v22.12.7 for isolates *B. coriaceae* Co53_lib1, *B. puertoricensis* SUM, *B. parkeri* SLO, and *B. turicatae* BTE5EL, and v23.04.3, for isolates *B. anserina* BA2, *B. hermsii* DAH, *B. hermsii* YOR, *B. coriaceae* Co53_lib3, *B. venezuelensis* RMA01, and *B. turicatae* 91E135. MinKNOW was supplied with the RefSeq version of each isolate's F27 plasmid (*B. anserina* BA2: NZ_CP073131.1, *B. hermsii* DAH: NZ_CP073141.1, *B. hermsii* YOR: NZ_CP073152.1, *B. coriaceae* Co53: NZ_CP075082.1, *B. puertoricensis* SUM: NZ_CP075390.1, *B. parkeri* SLO: NZ_CP073163.1, *B. venezuelensis* RMA01: NZ_CP073222.1, *B. turicatae* 91E135: NZ_CP073181.1, and *B. turicatae* BTE5EL: NZ_CP073197.1) for enrichment by adaptive sampling with an RTX 3060 Ti GPU enabled. The FAST5 data from MinKNOW were basecalled with Guppy (v6.5.7) using the high-accuracy model, a *Q*-score filter of 10, and --detect_mid_strand_adapter options. The basecalled data were demultiplexed using the same version of Guppy. Note that to have enough reads to analyze, the data of *B. coriaceae* lib1 and lib3 were processed individually, but their results were combined for read length analysis.

Basecalled data were analyzed to determine the read lengths of the DNA fragments mapping to the F27 plasmids in each condition. The basecalled data were first size selected using chopper v0.5.0 (110) to remove reads smaller than 700 bases, roughly the sequence length used for sampling reads by the MinKNOW software during adaptive sampling. The filtered data were mapped to their respective RefSeq genome assemblies via minimap2. The reads that mapped specifically to the F27 plasmid in each genome assembly (primary reads) were extracted by samtools with the -SbF 0×104 options with the RefSeq accession number of the specific F27 plasmid indicated. The resulting BAM file of primary reads mapped to the F27 plasmid was converted to FASTA file using samtools. Read lengths were calculated using an awk command and output as a TSV file. These read lengths were plotted for each condition for each isolate as scatterplots using RStudio (2023.03.0 Build 386, R version 4.3.0) with ggplot2 (v3.4.2) (111). All read length data including the total number of reads for each condition and the number of reads that mapped to the F27 plasmid for each condition can be found in Supplemental File 5. This file also indicates the SRA accession numbers used to generate the read length data for the F27 plasmids from the initial DNA sequencing data for genome assembly. Read counts were determined for the S1 and restriction enzyme double digested DNA samples by using the COUNTIFS function in Excel counting the read lengths for fragments A, B, and C for each isolate based on the reads between the predicted fragment length either plus or minus 5% of the predicted length. These data were plotted using RStudio and ggplot2 as well. Examples of the for-loops used to perform the above analysis as well as the commands for plotting the read lengths in R are available in Supplemental File 3. BAM file visualization of the F27 plasmid read mappings was done with IGV (v2.16.0) and annotated in Inkscape. In the IGV visualizations, indels <10 bp were masked, and reads were sorted by read order.

To confirm the presence of multiple 3′ ends in the F28 plasmids in *B. venezuelensis* RMA01, we performed adaptive sampling using the same genomic DNA that was sequenced to generate the RMA01 genome assembly. We used *B. turicatae* 91E135 as a negative control since the F28 plasmid's 3′ end should only be present in a single configuration. To increase the amount of available DNA ends for ligation for the ONT sequencing adapters and fragment the DNA for better adaptive sampling performance, we performed a restriction digest on the genomic DNA. A single BstXI site was present on the F28 plasmids within the PF50 plasmid partition gene, which is ~3 kb upstream of the *vmp* expression site. The genomic DNA was digested with BstXI (New England Biolabs) for 1 hour at 37°C with enzyme inactivation at 65°C per manufacturer's protocol. The genomic DNA was cleaned up with a 1:1 (vol/vol) NEBNext Sample Purification Beads with two 80% ethanol washes. The digested DNA was eluted into 10 µL of water and quantified using a Qubit 4 with the Qubit 1× dsDNA broad range assay kit.

The digested DNA was sequenced on ONT's MinION Mk1B platform. Two hundred fifty nanograms of digested DNA were end prepped and dA-tailed with the NEBNext Companion Module for ONT Ligation Sequencing. The prepared DNA was barcoded, and adapters were ligated from the SQK-NBD114.24 kit following manufacturer's instructions. Sixty nanograms of prepared DNA were loaded onto an R10.4.1 flow cell and sequenced using MinKNOW 22.10.7 with an RTX 3060 Ti enabled. Adaptive sampling was enabled to enrich for reads mapping to *B. venezuelensis* RMA01's lp35 (CP073229.1) and lp37 (CP073230.1) and *B. turicatae* 91E135's lp40 (CP073188.1) using a FASTA file containing the sequences for these plasmids. The FAST5 data were basecalled with Guppy (v6.4.2) using the super accurate model, a *Q*-score filter of 10, --detect_mid_strand_adapter, and --calib_detect options. The basecalled data were demultiplexed using the same version of Guppy.

Basecalled data were filtered and mapped to their respective genome assemblies to visualize read support for F28 plasmid 3′ ends. The data were filtered using NanoFilt (v2.8.0) -q 10 and -l 15,000. The filtered data were mapped to their respective genome assemblies (either *B. venezuelensis* RMA01 GCA_023035835.1 or *B. turicatae* 91E135 GCA_023035855.1) using minimap2. Primary reads were extracted using samtools (v1.11)

with the -SbF 0×104, and -q 20 options. The resulting BAM file for each plasmid was visualized using IGV and annotated in Inkscape. In the IGV visualizations, indels <10 bp were masked, supplementary reads were linked, and reads were sorted by read order.

## ACKNOWLEDGMENTS

We would like to thank Sebastián Muñoz Leal and Marcelo B. Labruna for providing us the *B. venezuelensis* RMA01 isolate used for adaptive sequencing.

This work was funded by NIH grants AI137412 (J.E.L.) and AI123652 (J.E.L.). The funders had no role in study design, data collection and interpretation, or the decision to submit the work for publication.

A.R.K. conceptualized the idea, performed the formal analysis and investigation, and wrote and edited the manuscript. J.E.L. conceptualized the idea, acquired the funding, supervised, and wrote and edited the manuscript.

## AUTHOR AFFILIATIONS

[1]Department of Pediatrics, Baylor College of Medicine, Houston, Texas, USA
[2]Department of Molecular Virology and Microbiology, Baylor College of Medicine, Houston, Texas, USA

## AUTHOR ORCIDs

Alexander R. Kneubehl ⓘ http://orcid.org/0000-0002-6243-4096
Job E. Lopez ⓘ http://orcid.org/0000-0003-4802-5779

## FUNDING

| Funder | Grant(s) | Author(s) |
| --- | --- | --- |
| HHS | NIH | National Institute of Allergy and Infectious Diseases (NIAID) | AI137412, AI123652 | Job E. Lopez |

## AUTHOR CONTRIBUTIONS

Alexander R. Kneubehl, Conceptualization, Data curation, Formal analysis, Methodology, Visualization, Writing – original draft, Writing – review and editing | Job E. Lopez, Conceptualization, Formal analysis, Funding acquisition, Supervision, Writing – original draft, Writing – review and editing

## DATA AVAILABILITY

Sequencing data generated by this study have been deposited to NCBI's Sequence Read Archive (SRA) and are available through BioProjects PRJNA989663 (F27 plasmid enzymatic digest adaptive sampling) and PRJNA918510 (adaptive sampling with *B. venezuelensis* RMA01 and *B. turicatae* 91E135 F28 plasmids).

## ADDITIONAL FILES

The following material is available online.

### Supplemental Material

**Supplemental figures (Spectrum00895-23-s0001.pdf).** Fig. S1 to S22.
**Supplemental file 1 (Spectrum00895-23-s0002.xlsx).** This file contains a sheet for each of the 5 repetitive blocks. The sheets contain the gene loci found in each of the 5 repetitive blocks for all the isolates analyzed.
**Supplemental file 2 (Spectrum00895-23-s0003.xlsx).** *bptA* gene analysis.
**Supplemental file 3 (Spectrum00895-23-s0004.txt).** Example bioinformatic commands.

**Supplemental file 4 (Spectrum00895-23-s0005.xlsx).** This file contains the loci classified as either a vsp or vlp for each of the isolates investigated. Vlp genes are further classified by subfamily.

**Supplemental file 5 (Spectrum00895-23-s0006.xlsx).** This file contains read lengths for the RBK library preparation initially used to generate the genome assemblies for the isolates analyzed, the read lengths for the circular and linear exemplars, and the read count and read length data for the F27 topology and size analysis.

## Open Peer Review

**PEER REVIEW HISTORY (review-history.pdf).** An accounting of the reviewer comments and feedback.

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
