## [Reviewer comments · Microbiology Spectrum]

Microbiology Spectrum

Comparative genomics analysis of three conserved plasmid families in the Western Hemisphere soft tick-borne relapsing fever borreliae provides insight into variation in genome structure and antigenic variation systems.

Alexander Kneubehl and Job Lopez

Corresponding Author(s): Job Lopez, Baylor College of Medicine

Review Timeline:

Submission Date:	March 8, 2023
Editorial Decision:	March 27, 2023
Revision Received:	July 20, 2023
Accepted:	July 24, 2023

Editor: Catherine Brissette

Reviewer(s): Disclosure of reviewer identity is with reference to reviewer comments included in decision letter(s). The following individuals involved in review of your submission have agreed to reveal their identity: Sherwood R Casjens (Reviewer #1); Brian Stevenson (Reviewer #2)

Transaction Report:

DOI: <https://doi.org/10.1128/spectrum.00895-23>

March 27, 2023

Dr. Job E. Lopez
Baylor College of Medicine
Pediatrics
1102 Bates avenue
Feigin C550
Houston, Texas 77030

Re: Spectrum00895-23 (Comparative genomics analysis of three conserved plasmid families in the Western Hemisphere soft tick-borne relapsing fever borreliae provides insight into variation in genome structure and antigenic variation systems.)

Dear Dr. Job E. Lopez:

Please address the mostly minor comments, although I do agree with Reviewer 2 about confirming that the F27 plasmids are linear by a secondary method.

Link Not Available

Sincerely,

Catherine Brissette

Journals Department
Reviewer comments:

Reviewer #1 (Comments for the Author):

This paper by Kneubehl and Lopez is a clearly written and long-awaited description of the universally present plasmids in the relapsing fever *Borrelia*'s. It is a significant step forward in this field. My specific comments are listed below.

1. Figure 1 - the strain name labels are very much too small. I had to expand the figures greatly on my computer screen to read them, and I suspect that they will be illegible without a magnifying glass in the journal.

2. line 133 - do the authors mean strain "B31"?
3. line 155 - the authors may be interested to know that bacteriophage tail fibers sometimes have collagen-like sequences (e.g., Smith et al. 1998 Bacteriophage collagen. Science 279: 1834).
4. Lines 168 & 227 - the paper is generally very clear and well written but I happened to notice a few places where the grammar needs improvement. Line 168 has a double negative and in line 227 "an" should be "a".
5. Sup Mat formatting could use some improvement (although this could be affected by the type of computer being used). Figure titles are often on the page preceding the figure itself, and fig S3 itself was missing on my screen. I recommend using PDF format which is more uniform than word processor formats.
6. line 172 and following paragraph
It is unclear to this reader whether or not any F27 plasmid DNA has been shown by direct experimentation to be "~10 kbp", or is this value deduced purely from sequencing information? The citation for this value is "reference 1", but it is unclear (to me at least) whether an experimental proof of the physical size is buried in that paper somewhere (which is a monumental paper describing/comparing various aspects of several whole genomes) - just showing that the strains have an "~10 kbp" plasmid in electrophoresis gels is not sufficient without proof that that gel band is indeed F27. Although, the authors are probably correct in their deduction of the inverted repeat structure of F17 plasmids, they should be clearer regarding this point, since an "ultimate proof" of this plasmid structure should include a physical measurement of its size.
The authors might wish to mention that Marconi et al. (1996 J. Bacteriol. 178: 3357) used Southern blots to show (before sequences were available) that *B. japonica* HO14 plasmid lp54 has the same overall inverted repeat structure as they propose for F27.
7. Figure 4 panel A - again the strain labels on the left are extremely small. In this case it will be difficult to make them larger without affecting the figure layout. I leave it to the authors to decide whether it is worthwhile to make the figure more accessible by making the strain labels larger in some way.
8. A more overt discussion of intra-species vs inter-species plasmid diversity would be a welcome addition to this paper.

Reviewer #2 (Public repository details (Required)):

Needs reference to *Borrelia coriaceae* genome sequence

Reviewer #2 (Comments for the Author):

Kneubehl and Lopez present a comparative analysis of three conserved plasmids found in isolates of western hemisphere relapsing fever *Borrelia*. This is an interesting and useful report that will be widely appreciated by the field.

As much as I dislike requesting additional experiments, in this case, some are necessary. The F27 family of plasmids has been reported to be either circular or linear. The authors claim (lines 171-201) that they must be linear, based on their sequencing analyses. I am not convinced. A simple, straightforward method to test their hypothesis would be PCR: primers oriented outwards from each "telomere" will not give a product if the plasmid is linear, but will if the plasmid is circular, while primers oriented inwards from each "telomere" will give products regardless of structure. This should be done for each strain, as it was previously demonstrated that the "cp32-like" plasmids of RF *Borrelia* can be either circular or linear, depending upon the *Borrelia* species. A much better method would be Southern blotting/restriction mapping, but that might be too much to ask nowadays.

Related to the above, lines 362-365 refer to studies of F27 plasmids using pulsed-field electrophoresis. That method does not resolve circular plasmids. If a circular plasmid is cut/sheared, that will resolve and give the appearance of a normally linear plasmid.

Other comments:

- 1) Line 19: since several species are being referred to, this should be "have complex genomes" (plural genome).
- 2) Line 25: please clarify, "syntenic across all examined relapsing fever species". The authors cannot claim synteny across all species, since they did not look at all of them
- 3) Lines 54-55: the phrase "plasmid resolved genomes" is confusing. I assume that the authors are referring to genome sequence assemblies, rather than the actual genomes. Please clarify this. There are other places in the manuscript where the authors use "genome" as jargon to stand in for "genome assembly" or "genome sequence" - please re-read and revise as

warranted.

4) Lines 88-91 and elsewhere: the authors italicize protein names unnecessarily. Only gene names should be italicized. Fixing this now will help a copy editor down the line :)

5) Lines 195-201: The absence of a PF50 gene from the F27 plasmid family is reminiscent of the cp9 of *B burgdorferi* B31 and other small circular plasmids of *Lyme borreliæ*. cp9 plasmids are also largely devoid of genes of known function. Please discuss this, as well as differences from the *Lyme* cp9 plasmids.

6) Lines 222-224: Have the UHS and DHS regions been conclusively demonstrated to assist in recombination, or is this still an unproven hypothesis? If not yet demonstrated to be a fact, then please change "identified" to "hypothesized".

7) Lines 245-251: *B coriaceæ* is mentioned here and in the figures, but I do not see a reference to its genome sequence. Please provide a sequence reference.

8) Lines 401-404: This is the only place mention that the studied *B anserina* strain has only a single vmp gene. This is very important to know, and should be prominently stated in the results section, as well as having its own paragraph in the discussion.

Staff Comments:

Preparing Revision Guidelines

Please return the manuscript within 60 days; if you cannot complete the modification within this time period, please contact me. If you do not wish to modify the manuscript and prefer to submit it to another journal, please notify me of your decision immediately so that the manuscript may be formally withdrawn from consideration by Microbiology Spectrum.

We are grateful that the reviewers appreciated the work in this manuscript. We appreciated their thoughtful comments and critiques. We have added more data that further indicated the F27 plasmids are linear using a restriction digest/adaptive sampling sequencing strategy (additional figures and text discussed below). We have made the changes requested in the point-by-point response below and feel that these strengthen the manuscript and increase clarity.

Reviewer comments:

Reviewer #1 (Comments for the Author):

This paper by Kneubehl and Lopez is a clearly written and long-awaited description of the universally present plasmids in the relapsing fever *Borrelia*'s. It is a significant step forward in this field. My specific comments are listed below.

Reviewer 1 Comment 1: Figure 1 - the strain name labels are very much too small. I had to expand the figures greatly on my computer screen to read them, and I suspect that they will be illegible without a magnifying glass in the journal.

Authors' Response: Thank you, we have added strain designations to the left of each plasmid in that alignment and have indicated that in Figure 1's caption as well, see lines 816 through 819 in the tracked changes copy.

Reviewer 1 Comment 2: line 133 - do the authors mean strain "B31"?

Authors' Response: Indeed, we did mean B31. Thank you for the correction. This has been corrected in the text at line 142 in the tracked changes version.

Reviewer 1 Comment 3: line 155 - the authors may be interested to know that bacteriophage tail fibers sometimes have collagen-like sequences (e.g., Smith et al. 1998 Bacteriophage collagen. Science 279: 1834).

Authors' Response: Thank you for bringing this to our attention. We are currently working on identifying the phage-related plasmids/genes in the Western Hemisphere soft tick-borne relapsing fever genomes dataset and will incorporate this into future analyses.

Reviewer 1 Comment 4: Lines 168 & 227 - the paper is generally very clear and well written but I happened to notice a few places where the grammar needs improvement. Line 168 has a double negative and in line 227 "an" should be "a".

Authors' Response: Thank you. We have corrected the double negative on lines 178-179 in tracked changes version to "Further, both BLASTn and BLASTp did not..." and the article used in line 227 to "Since *B. turicatae* utilizes a *B. burgdorferi* ..." on line 359 in the tracked changes version.

Reviewer 1 Comment 5: Sup Mat formatting could use some improvement (although this could be affected by the type of computer being used). Figure titles are often on the page preceding the figure

itself, and fig S3 itself was missing on my screen. I recommend using PDF format which is more uniform than word processor formats.

Authors' Response: We agree that using a PDF format would more appropriate and have made that change.

Reviewer 1 Comment 6: line 172 and following paragraph

It is unclear to this reader whether or not any F27 plasmid DNA has been shown by direct experimentation to be "~10 kbp", or is this value deduced purely from sequencing information? The citation for this value is "reference 1", but it is unclear (to me at least) whether an experimental proof of the physical size is buried in that paper somewhere (which is a monumental paper describing/comparing various aspects of several whole genomes) - just showing that the strains have an "~10 kbp" plasmid in electrophoresis gels is not sufficient without proof that that gel band is indeed F27. Although, the authors are probably correct in their deduction of the inverted repeat structure of F17 plasmids, they should be clearer regarding this point, since an "ultimate proof" of this plasmid structure should include a physical measurement of its size.

The authors might wish to mention that Marconi et al. (1996 J. Bacteriol. 178: 3357) used Southern blots to show (before sequences were available) that *B. japonica* HO14 plasmid Ip54 has the same overall inverted repeat structure as they propose for F27.

Authors' Response: We agree with the concerns of both reviewers regarding the topology of the F27 plasmids. We elected to use a modified sequencing strategy to determine the topology of these plasmids. Nanopore sequencing determines the sequence of a DNA fragment while at the same time measures the length of the fragment (when using the ligation-based library preparation protocol approach). We leveraged both aspects of this sequencing platform to sequence restriction digested genomic DNA from each isolate. The restriction enzyme for each isolate was chosen such that the fragmentation pattern of the F27 plasmid would be diagnostic of its topology (see Figure 3). We also incorporated use of the S1 nuclease which can cleave the ssDNA telomeres of linear plasmids in *Borrelia*. When S1 nuclease and a restriction enzyme are used together with adaptive sampling to enrich reads for the F27 plasmids, the sequencing data reflected the fragmentation pattern we would predict for a linear plasmid of the size we reported for each isolate. These data supported our hypothesis that these plasmids are indeed linear in topology as well as being large, inverted repeats. The additional data and associated manuscript text are found on the following lines in the tracked changes version 193-200, 212-333, discussed on lines 501-526, with methods indicated on lines 693-765 with Figure 3, Figures S3-S13 added.

Reviewer 1 Comment 7: Figure 4 panel A - again the strain labels on the left are extremely small. In this case it will be difficult to make them larger without affecting the figure layout. I leave it to the authors to decide whether it is worthwhile to make the figure more accessible by making the strain labels larger in some way.

Authors' Response: We increased the size of panel A strains appear clearer. When viewed digitally it should be easier to zoom in on the panel if need be.

Reviewer 1 Comment 8: A more overt discussion of intra-species vs inter-species plasmid diversity would be a welcome addition to this paper.

Authors' Response: We agree, but there are limited numbers of plasmid-resolved genome assemblies within each species. We have discussed the differences between species in our first paper with this dataset (<https://doi.org/10.1186/s12864-022-08523-7>) and to some extent intra-specific differences between *B. turicatae* 91E135 and BTE5EL. Work is on-going to sequence our collection of *B. turicatae* isolates to perform a more in-depth intra-specific comparison of that species. Though more work needs to be done across all relapsing fever species.

Reviewer #2 (Comments for the Author):

Reviewer #2 (Public repository details (Required)):

Needs reference to *Borrelia coriaceae* genome sequence

Kneubehl and Lopez present a comparative analysis of three conserved plasmids found in isolates of western hemisphere relapsing fever *Borrelia*. This is an interesting and useful report that will be widely appreciated by the field.

As much as I dislike requesting additional experiments, in this case, some are necessary. The F27 family of plasmids has been reported to be either circular or linear. The authors claim (lines 171-201) that they must be linear, based on their sequencing analyses. I am not convinced. A simple, straightforward method to test their hypothesis would be PCR: primers oriented outwards from each "telomere" will not give a product if the plasmid is linear, but will if the plasmid is circular, while primers oriented inwards from each "telomere" will give products regardless of structure. This should be done for each strain, as it was previously demonstrated that the "cp32-like" plasmids of RF *Borrelia* can be either circular or linear, depending upon the *Borrelia* species. A much better method would be Southern blotting/restriction mapping, but that might be too much to ask nowadays.

Related to the above, lines 362-365 refer to studies of F27 plasmids using pulsed-field electrophoresis. That method does not resolve circular plasmids. If a circular plasmid is cut/sheared, that will resolve and give the appearance of a normally linear plasmid.

Authors' Response: Please see response to **Reviewer 1 Comment 6**. We addressed this concern by implementing an adaptive sequencing approach (details are provided to the response of Reviewer 1 Comment 6).

Other comments:

Reviewer 2 Comment 1: Line 19: since several species are being referred to, this should be "have complex genomes" (plural genome).

Authors' Response: We have corrected the text to pluralize genomes on line 20 of the tracked changes.

Reviewer 2 Comment 2: Line 25: please clarify, "syntenic across all examined relapsing fever species". The authors cannot claim synteny across all species, since they did not look at all of them.

Authors' Response: We have incorporated the suggested text in lines 26-27 in the tracked changes version to "syntenic across all the RF *Borrelia* species that we examined..." to better clarify that we mean the Western Hemisphere soft tickborne relapsing fever species.

Reviewer 2 Comment 3: Lines 54-55: the phrase "plasmid resolved genomes" is confusing. I assume that the authors are referring to genome sequence assemblies, rather than the actual genomes. Please clarify this. There are other places in the manuscript where the authors use "genome" as jargon to stand in for "genome assembly" or "genome sequence" - please re-read and revise as warranted.

Authors' Response: Thank you for bringing this to our attention. We have changed "plasmid-resolved genomes" to "plasmid-resolved genome assemblies" on lines 24, 41, 58, 360, 372, 378, 415, 448, 611.

Reviewer 2 Comment 4: Lines 88-91 and elsewhere: the authors italicize protein names unnecessarily. Only gene names should be italicized. Fixing this now will help a copy editor down the line :)

Authors' Response: Thank you for bringing this to our attention. We have corrected the italicization of protein names throughout the manuscript.

Reviewer 2 Comment 5: Lines 195-201: The absence of a PF50 gene from the F27 plasmid family is reminiscent of the cp9 of *B. burgdorferi* B31 and other small circular plasmids of Lyme borreliae. cp9 plasmids are also largely devoid of genes of known function. Please discuss this, as well as differences from the Lyme cp9 plasmids.

Authors' Response: From an analysis of the cp9 plasmids, they are occasionally missing the PF32 genes but have the PF49, PF50, and PF57/62 genes (<https://doi.org/10.1186/s12864-017-3553-5>, <https://doi.org/10.1371/journal.pone.0033280>, and [10.1046/j.1365-2958.2001.02256.x](https://doi.org/10.1046/j.1365-2958.2001.02256.x)). The cp9 plasmids also are not related to the F27 plasmids based on the PF57/62 and PF32 phylogenetic analysis we previously performed (<https://doi.org/10.1186/s12864-022-08523-70>). Many of the genes in spirochete genomes are genes of unknown function. Based on the differences we have noted and the additional length of the manuscript, there is not enough similarity between the cp9s and the F27s to warrant further analysis in the current work. We hope this is acceptable.

Reviewer 2 Comment 6: Lines 222-224: Have the UHS and DHS regions been conclusively demonstrated to assist in recombination, or is this still an unproven hypothesis? If not yet demonstrated to be a fact, then please change "identified" to "hypothesized".

Authors' Response: We have changed "identified" to "hypothesized" on line 355 in the tracked changes, as we were not able to find in the literature that the sites have been experimentally confirmed to assist in recombination.

Reviewer 2 Comment 7: Lines 245-251: *B. coriaceae* is mentioned here and in the figures, but I do not see a reference to its genome sequence. Please provide a sequence reference.

Authors' Response: Thank you for catching this. We have added the GenBank accession for the *B. coriaceae* Co53 genome assembly used in this work in line 627 in the tracked changes version.

Reviewer 2 Comment 8: Lines 401-404: This is the only place mention that the studied *B. anserina* strain has only a single vmp gene. This is very important to know, and should be prominently stated in the results section, as well as having its own paragraph in the discussion.

Authors' Response: We agree this is a finding worth expanding upon. We have explicitly indicated this in the results section (lines 415-416 of the tracked changes version) and have expanded discussion of this into its own paragraph in the discussion (lines 565-590 of the tracked changes version).

July 24, 2023

Dr. Job E. Lopez
Baylor College of Medicine
Pediatrics
1102 Bates avenue
Feigin C550
Houston, Texas 77030

Re: Spectrum00895-23R1 (Comparative genomics analysis of three conserved plasmid families in the Western Hemisphere soft tick-borne relapsing fever borreliae provides insight into variation in genome structure and antigenic variation systems.)

Dear Dr. Job E. Lopez:

Your manuscript has been accepted, and I am forwarding it to the ASM Journals Department for publication. You will be notified when your proofs are ready to be viewed.

Sincerely,

Catherine Brissette
Editor, Microbiology Spectrum
